# Dissipative dynamics in the free massive boson limit of the sine-Gordon model

Ádám Bácsi,[1, 2, *] Cătălin Paşcu Moca,[3, 4] Gergely Zaránd,[3, 5] and Balázs Dóra[1, 6]

[1]*MTA-BME Lendület Topology and Correlation Research Group,*
*Budapest University of Technology and Economics, 1521 Budapest, Hungary*
[2]*Department of Mathematics and Computational Sciences,*
*Széchenyi István University, 9026 Győr, Hungary*
[3]*MTA-BME Quantum Dynamics and Correlations Research Group,*
*Budapest University of Technology and Economics, 1521, Budapest, Hungary*
[4]*Department of Physics, University of Oradea, 410087, Oradea, Romania*
[5]*BME-MTA Exotic Quantum Phases Research Group, Department of Theoretical Physics,*
*Budapest University of Technology and Economics, Budapest, Hungary*
[6]*Department of Theoretical Physics, Budapest University of Technology and Economics, Budapest, Hungary*

(Dated: November 24, 2021)

We study the dissipative dynamics of one-dimensional fermions, described in terms of the sine-Gordon model in its free massive boson or semi-classical limit, while keeping track of forward scattering processes. The system is prepared in the gapped ground state, and then coupled to environment through local currents within the Lindblad formalism. The heating dynamics of the system is followed using bosonization. The single particle density matrix exhibits correlations between the left and right moving particles. While the density matrix of right movers and left movers is translationally invariant, the left-right sector is not, corresponding to a translational symmetry breaking charge density wave state. Asymptotically, the single particle density matrix decays exponentially with exponent proportional to $-\gamma t|x|\Delta^2$ where $\gamma$ and $\Delta$ are the dissipative coupling and the gap, respectively. The charge density wave order parameter decays exponentially in time with an interaction independent decay rate. The second Rényi entropy grows linearly with time and is essentially insensitive to the presence of the gap.

## I. INTRODUCTION

Interacting quantum particles in one dimension typically either retain some low energy excitations and form a Luttinger liquid, or become gapped due to strong interactions and realize, e.g., an insulator. The latter phase is often described by the famous sine-Gordon model [1, 2] or by its close relative, the simpler free massive boson model. The behavior of these systems is well documented in closed quantum systems [3–6] in and out of equilibrium [7, 8], and faithfully realized in condensed matter, cold atom or other complex systems. In spite of its relevance, the response of these models to dissipative coupling to environment is much less studied and understood.

The interplay of dissipation and strong correlations, combined with reduced dimensionality [9–18], promises to provide a plethora of interesting phenomena [19–26]. This includes algebraic or exponential decay of correlation function in dissipative many-particle systems[10], universal features in the vaporization dynamics of Luttinger liquids[27] or targeted cooling into topological states from arbitrary initial states[19, 26]. In addition, propagators and correlation spreading also reveal unexpected features [14]. A recent experiment [28] on Josephson-coupled one dimensional bosons as well as related theoretical analysis [29] also hint to the importance of incorporating dissipation into sine-Gordon the-

ory, though other explanations are also available[30, 31].

This motivates us to take a closer look at the dissipative dynamics of the sine-Gordon (sG) model. The sG model is Bethe ansatz solvable as a closed quantum system [8, 32], its solution simplifies, however, significantly at the so-called Luther-Emery line [8, 32] and also in the semi-classical [33–36] or free massive boson limit, when solitons are neglected, and the interaction potential is expanded around one of its minima, reducing the model to the massive boson or Klein-Gordon model [33]. This 'free massive boson' limit captures the basic correlations, and we focus on this limit to understand dissipative dynamics.

We consider spinless interacting fermions in the presence of a gap in their spectrum, arising due to backscattering or umklapp processes [8], or simply due to a staggered dimerizing deformation or potential of a one dimensional half-filled lattice. We assume a dissipation coupled to the current operator, and turn on dissipation at time $t = 0$, as a certain kind of *dissipative quantum quench*. We use bosonization to investigate the spatial and temporal decay of the fermionic single particle density matrix

$$G_{\alpha\beta}(x;t) \equiv \langle\psi_\alpha^+(x)\psi_\beta(0)\rangle , \qquad \alpha,\beta = L,R ,$$

with $\psi_L$ and $\psi_R$ referring to the left- and right moving fermionic fields (see Eq. (10)), respectively and the expectation value defined as $\langle...\rangle = \text{Tr}\{\rho(t)...\}$, with $\rho(t)$ the time evolved density operator. We write the single particle density matrix as

$$G_{\alpha\beta}(x;t) = G_{\alpha\beta}(x;0)\,e^{F_{\alpha\beta}(x,t)} , \qquad (1)$$

* bacsi.adam@sze.hu

and determine the functions $F_{RR}(x, t)$ and $F_{LR}(x, t)$. In contrast to the dissipative Luttinger liquid case [27], the correlator between right and left moving fermions becomes finite due to the presence of the gap.

We find that in the long distance and long time limit, the single particle density matrices of both $G_{RR}$ and $G_{LR}$ decay exponentially with an exponent proportional to $-\gamma t |x| \Delta^2$ where $\gamma$ and $\Delta$ are the dissipative coupling and the gap, respectively. The von Neumann entropy of the system also grows similarly to the Luttinger liquid as $-t \ln(t)$, and is largely insensitive to the presence of the gap. These results are also relevant for dissipative interacting relativistic field theories, such as the massless Thirring model[1, 2].

## II. FREE MASSIVE BOSON LIMIT OF THE SINE-GORDON MODEL

The one dimensional sine-Gordon (sG) model [1, 6, 8, 32] describes interacting fermions, bosons or spins [8]. The sG model is constructed in terms of the Bose field, $\phi(x)$ [8],

$$\phi(x) = -i \sum_{q \neq 0} \sqrt{\frac{\pi |q|}{2L}} \frac{1}{q} e^{-iqx} \left( b_q^+ + b_{-q} \right) \quad (2)$$

with $b_q$ denoting canonical bosonic operators of momentum $q$. It consists of the conventional quadratic Luttinger liquid Hamiltonian, $H_{LL}$, incorporating forward scattering processes, and an additional $-J \int dx \cos(2\phi(x))$ perturbation. If the cosine term is relevant, it opens up a gap in the spectrum, and the field $\phi(x)$ gets locked into its minima.

This cosine term can arise due to strong umklapp scattering [8] or backscattering [2], or simply due to a staggered potential or dimerization [37], which can all open up a gap in the spectrum. For the sake of concreteness, we can consider a one dimensional tight binding, half filled lattice of spinless fermions, interacting via nearest neighbour interaction, in the presence of a staggered potential [8]. This maps onto the anisotropic XXZ Heisenberg chain in the presence of a staggered magnetic field in the $z$ direction [5]. Strong repulsive interaction without the staggered potential, or weak interactions in the presence of the staggered potential yield the sine-Gordon type low energy physics. Alternatively, one can consider a one dimensional p-wave superconductor in the presence of electron-electron interaction [38], though the cosine term would contain the dual field of $\phi(x)$.

Deep in the massive phase, one can approximate the cosine term as $\sim \int dx 2J\phi^2(x)$ [33]. This represents the so-called semi-classical limit of the model, which is described by massive bosonic excitations [33]. In this limit, the Hamiltonian can be written in terms of the bosonic annihilation operators $b_q$ as

$$H = \sum_{q>0} \left[ \omega(q) \left( b_q^+ b_q + b_{-q}^+ b_{-q} \right) + g(q) \left( b_q^+ b_{-q}^+ + b_q b_{-q} \right) \right]. \quad (3)$$

Here, $\omega(q) = (v_0 + g_4)q + \Delta^2/(2v_0 q)$ and $g(q) = g_2 q + \Delta^2/(2v_0 q)$ with $v_0$ the bare sound velocity of the non-interacting and non-gapped system, $g_2$ and $g_4$ are the conventional forward scattering interactions[8, 39] and the energy scale $\Delta \sim 2J$ is proportional to the gap or mass of the elementary excitations. Note that in the absence of $\Delta$, the model is identical to the interacting Luttinger model [27].

Since the Hamiltonian is quadratic in the bosonic operators, it can be diagonalized by Bogoliubov transformation leading to

$$H = E_{GS} + \sum_{q>0} \tilde{\omega}(q) \left( B_q^+ B_q + B_{-q}^+ B_{-q} \right), \quad (4)$$

where $\tilde{\omega}(q) = \sqrt{\omega^2(q) - g^2(q)} = \sqrt{(\tilde{v}|q|)^2 + \tilde{\Delta}^2}$ is the gapped spectrum, and $E_{GS} = \sum_{q>0}(\tilde{\omega}(q) - \omega(q))$ is the ground state energy. Due to the interaction, the sound velocity and the gap are renormalized as $\tilde{v} = \sqrt{(v_0 + g_4)^2 - g_2^2}$ and $\tilde{\Delta} = \Delta\sqrt{K\tilde{v}/v_0}$, where $K = \sqrt{(v_0 + g_4 - g_2)/(v_0 + g_4 + g_2)}$ is the Luttinger parameter. For attractive or repulsive interactions, $K > 1$ or $K < 1$, respectively.

The gap $\tilde{\Delta}$ and the renormalized sound velocity, $\tilde{v}$, define a natural length scale, the coherence length,

$$\tilde{\xi} = \tilde{v}/\tilde{\Delta}. \quad (5)$$

As we shall see later, the coherence length $\tilde{\xi}$ represents a characteristic length scale that separates the short and the long distance regimes for the spatial dependence of the fermionic Green's functions $G(x; t)$.

We note that this semi-classical limit is expected to be valid as long as the Luttinger liquid parameter is small, $K \ll 1$. In this case, expanding the cosine potential around one of its minima is a rather good approximation[8, 33–36]. In this limit, the mass of the soliton is pushed up[1] to very high energies with $1/\sqrt{K}$ and thus is expected to decouple from the dynamics. Its effect can still be taken into account perturbatively using a form factor expansion, which is beyond the scope of the current investigation.

The goal of the present investigation is to study the non-equilibrium dynamics of the semi-classical limit of the sine-Gordon model following a sudden quench at $t = 0$ when the coupling to the environment is switched on. For the dynamics, the initial state is the ground state of Eq. (4), i.e., no bosonic excitations are present. After the quench, $t > 0$, the time evolution of the system is governed by the quantum master equation of Lindblad type for the density operator

$$\partial_t \rho = -i[H, \rho] + \gamma \int dx \left( [j(x), \rho j(x)] + h.c. \right) \quad (6)$$

where $\rho$ is the density operator. In Eq. (6) $j(x)$ is the current operator, which is a natural choice of jump operator when the environment is a fluctuating vector potential or gauge field [12, 27, 40–43]. These naturally couple to local currents. We mention that another natural choice for the jump operator would be the local particle density[44–46]. Our theory would in principle apply also for that case with minor modifications, e.g. by replacing $K$ with $1/K$. In the bosonization language, the current operator is expressed with the help of the bosonic annihilation operators [8] as

$$ j(x) = \sum_{q \neq 0} \sqrt{\frac{|q|}{2\pi L}} \, \mathrm{sgn}(q) \, e^{-iqx} \left( b_{-q} - b_q^+ \right) , \quad (7) $$

and the integral in Eq. (6) yields

$$ \partial_t \rho = -i[H, \rho] + \frac{\gamma}{2\pi} \sum_{q \neq 0} \left( [L_q, \rho L_q^+] + h.c. \right) \quad (8) $$

where $L_q = \sqrt{|q|} \left( b_q - b_{-q}^+ \right) = \sqrt{|q|} \left( B_q - B_{-q}^+ \right) / \sqrt{\mathcal{K}(q)}$ with $\mathcal{K}(q) = K\tilde{v}|q|/\tilde{\omega}(q)$.

The Lindblad equation allows us to calculate the expectation value of the occupation number of Bogoliubov bosons $B_q^+ B_q$ and the anomalous operator, which are obtained analytically as

$$ n_q^B(t) = \mathrm{Tr}\left\{ \rho(t) B_q^+ B_q \right\} = \frac{\gamma}{\pi \tilde{v} K} \tilde{\omega}(q) t \quad (9a) $$

$$ m_q^B(t) = \mathrm{Tr}\left\{ \rho(t) B_q^+ B_{-q}^+ \right\} = \frac{\gamma}{2i\pi \tilde{v} K} \left( e^{2i\tilde{\omega}(q)t} - 1 \right) . \quad (9b) $$

The linear growth of the boson number implies that in the long time limit the system is heated up to infinite temperature. This feature is the same as found in the gapless Luttinger model [27, 46], and follows from the fact that the jump operator is hermitian.

Having discussed the properties of the bosonic operators, we return now to the original fermionic fields, which are probed experimentally. The physical fermions are decomposed into right and left going particles as

$$ \psi(x) = e^{ik_F x} \psi_R(x) + e^{-ik_F x} \psi_L(x) , \quad (10) $$

where $k_F$ is the Fermi wavenumber and $\psi_{R,L}(x)$ is the field operator for the right $(R)$ and left $(L)$ moving fermionic quasiparticles. These can be expressed in terms of the bosons as

$$ \psi_{R/L} = \frac{1}{\sqrt{2\pi\alpha}} e^{i(\theta(x) \pm \phi(x))} , \quad (11) $$

with $\theta(x)$ the dual field $\phi(x)$ [8]. The relevance of the cosine term therefore implies the locking of the phase $\phi$, corresponding to a finite expectation value of $\psi_L^+ \psi_R$, and a spatially modulated particle density.

The equal time single particle density matrix is defined as

$$ \mathcal{G}(x, y; t) = \langle \psi^+(x)\psi(y) \rangle = $$
$$ = e^{ik_F(x+y)} \langle \psi_L^+(x)\psi_R(y) \rangle + c.c. + $$
$$ + e^{ik_F(y-x)} \langle \psi_R^+(x)\psi_R(y) \rangle + e^{ik_F(x-y)} \langle \psi_L^+(x)\psi_L(y) \rangle. $$

Here, the expressions within the expectation values are translational invariant, i.e. depend only on $x - y$, while the exponential prefactors $\exp(\pm ik_F(x+y))$ are not. In a Luttinger liquid without the sine-Gordon term, only the $RR$ and $LL$ correlators are finite, the $RL$ and $LR$ terms vanish identically. However, with the sine-Gordon term, these become also finite and need to be considered on equal footing.

The diagonal of the density matrix, $\mathcal{G}(x, x; t)$, is just the particle density,. As stated earlier, for any finite $\langle \psi_R^+(x)\psi_L(x) \rangle \neq 0$ it contains spatially inhomogeneous part $\sim e^{i2k_F x} \langle \psi_L^+(x)\psi_R(x) \rangle + c.c$, and describes a charge density wave pattern.

## III. EQUAL-TIME SINGLE PARTICLE DENSITY MATRIX OF THE RIGHT MOVERS

To have a deeper understanding about the heat up process of the fermions, it is worth studying the time evolution of the correlations. We investigate first the time dependence of the equal-time single particle density matrix of the right movers

$$ G_{RR}(x; t) = \mathrm{Tr}\left\{ \rho(t) \psi_R^+(x)\psi_R(0) \right\} , \quad (12) $$

with $\psi_R$ defined in Eq. (11). Later, we will also study the correlation between right- and left moving particles.

After a long algebraic derivation similar, for details see the Appendix, the single particle density matrix is calculated as

$$ \ln\left( \frac{G_{RR}(x; t)}{G_0(x)} \right) = -\sum_{q>0} \frac{8\pi}{Lq} n_q^b(t) \sin^2\left( \frac{qx}{2} \right), \quad (13) $$

where $G_0(x) = \frac{1}{2\pi(\alpha - ix)}$ is the single-particle density matrix of the gapless, non-interacting system, and $n_q^b(t) = \langle b_q^+ b_q \rangle$ is the occupation number of the $b$ bosons. The short length scale $\alpha$, appearing in $G_0$ and (11), is introduced to regularize the momentum integral by $e^{-\alpha|q|}$. The asymptotic behavior of the non-interacting gapless correlation function obeys the well-known $\sim 1/x$ power-law for $x \gg \alpha$. In our model, the single particle density matrix deviates from $G_0(x)$ due to the presence of both interaction and dissipation. The expectation value of the number of $b$-bosons is calculated as

$$ n_q^b(t) = \frac{\omega(q)}{\tilde{\omega}(q)} n_q^B(t) - \frac{g(q)}{\tilde{\omega}(q)} \mathrm{Re}\left( m_q^B(t) \right) + \frac{1}{2}\left( \frac{\omega(q)}{\tilde{\omega}(q)} - 1 \right), \quad (14) $$

in which the time dependence only occurs through the functions $n_q^B(t)$ and $m_q^B(t)$, defined in Eqs. (9). Here, the first two terms arise from dissipation, while the last one follows from quantum fluctuations due to interactions.

At $t = 0$, i.e., when the system is in the vacuum of $B$-bosons, both $n_q^B$ and $m_q^B$ vanish, and the initial single

particle density matrix is written as

$$G_{RR}(x;0) = \frac{i}{2\pi\alpha} \exp\left( -\sum_{q>0} \frac{4\pi}{L|q|} \frac{\omega(q)}{\tilde{\omega}(q)} \sin^2\left(\frac{qx}{2}\right) \right).$$
$$(15)$$

We use again the exponential cut-off with $\alpha$ to regularize the momentum integrals. For a small gap, $\tilde{\Delta} \ll \tilde{v}/\alpha$, i.e. $\alpha \ll \tilde{\xi}$, the $t = 0$ single-particle density matrix can be calculated analytically as

$$G_{RR}(x;0) = \frac{i}{2\pi\alpha} \exp\left( \frac{1}{2K}\left(1 + \frac{1}{2}G_M\left(\frac{x^2}{4\tilde{\xi}^2}\right)\right) + \right.$$
$$\left. + \frac{K+K^{-1}}{2}\left(\ln\left(\frac{\alpha}{\tilde{\xi}}\right) - \frac{\pi}{2}\mathrm{Re}\, Y_0\left(\frac{ix}{\tilde{\xi}}\right)\right)\right) \quad (16)$$

where $G_M(y) = G_{13}^{21}\left(y \,\middle|\, \begin{matrix} & 3/2 & \\ 0 & 1 & 1/2 \end{matrix}\right)$ is a Meijer G-function[47] and $Y_0$ the Bessel function of the second kind. It is meaningful to extract the short and long distance behaviors, obtained as

$$G_{RR}(x;0) = \frac{i}{2\pi\alpha} \begin{cases} \left|\frac{\alpha}{x}\right|^{(K+K^{-1})/2}, & \alpha \ll x \ll \tilde{\xi}, \\ \left(\frac{\alpha}{\tilde{\xi}}\right)^{(K+K^{-1})/2} e^{-\frac{\pi}{4K}\left|\frac{x}{\tilde{\xi}}\right|}, & \tilde{\xi} \ll x. \end{cases}$$
$$(17)$$

For short distances, $x \ll \tilde{\xi}$, the gap has no effect on the spatial decay and the power-law function with the exponent $(K + K^{-1})/2$ is characteristic of the interacting, gapless Luttinger model. At long distances, $x \gg \tilde{\xi}$, however, the gap becomes relevant and the correlation function decays exponentially with $x$.

After switching on the coupling to the environment, the single particle density matrix is expected to decrease with time according to the intuition that the system is heated up to infinite temperature. The time-dependent part of the equal-time correlation function is obtained by substituting the time-dependent terms of Eqs. (14) into (13),

$$F_{RR}(x,t) \equiv \ln\left(\frac{G_{RR}(x;t)}{G_{RR}(x;0)}\right) = \quad (18)$$
$$= -\sum_{q>0} \frac{8\pi}{Lq}\left[\frac{\omega(q)}{\tilde{\omega}(q)}n_q^B(t) - \frac{g(q)}{\tilde{\omega}(q)}\mathrm{Re}\left(m_q^B(t)\right)\right]\sin^2\left(\frac{qx}{2}\right).$$

The integral over $q$ can be calculated analytically only for the term containing $n_q^B(t)$. Using the exponential cut-off and taking the scaling limit, when $\alpha \ll \{x, \tilde{\xi}\}$, the integral yields the first term of Eq. (19), whose magnitude increases linearly with time.

The term containing $m_q^B(t)$ cannot be calculated analytically. Numerical investigation shows that this term features temporal oscillations for late times. These oscillations are captured analytically by approximating $\tilde{\omega}(q) \approx \tilde{\Delta}$. Together with the contribution of the $n_q^B(t)$

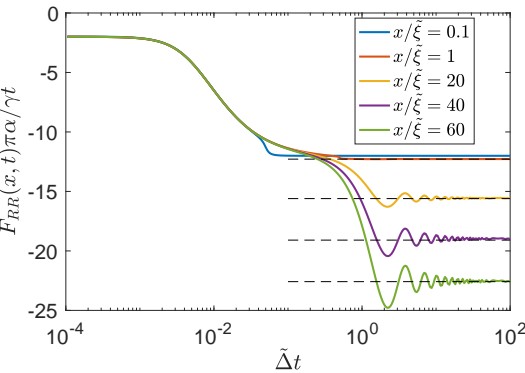

FIG. 1. Normalized effective decay rate of correlations in the $RR$ sector as a function of time at various distances for $K = 0.3$ and $\alpha/\tilde{\xi} = 0.01$. The curves are obtained by evaluating Eq. (18) numerically. They start from $-2$ in our dimensionless units for short times and reach the $x$ dependent asymptotic values (thin black dashed lines) from Eq. (20). In between, the crossover is described by Eq. (22).

term, we get in the $\tilde{\Delta}^{-1} \ll t$ long time limit

$$F_{RR}(x,t) \approx -\frac{\gamma t}{\pi\alpha}\left(\frac{\pi\alpha|x|}{2\tilde{\xi}^2 K^2} + 1 + \frac{1}{K^2}\right) +$$
$$+ \frac{\gamma}{\tilde{\xi}K^2}\frac{\sin(2\tilde{\Delta}t)}{2\tilde{\Delta}}\left(\left|\frac{x}{2\tilde{\xi}}\right| + \frac{\tilde{\xi}}{\pi\alpha}\left(1 - K^2\right)\right). \quad (19)$$

For long times, the first term displaying the linear time dependence dominates in Eq.(19), and yields an exponential decay of the density matrix, the exponent of which decreases linearly with increasing separation $x$ (see Fig. 1). For times $\tilde{\Delta}^{-1} \lesssim t$ the second term yields oscillations of the exponent, but these oscillations fade away at longer times, and an exponential suppression with exponent

$$F_{RR}(x,t) \approx -\frac{\gamma t}{\pi\alpha}\left(\frac{\pi\alpha|x|}{2\tilde{\xi}^2 K^2} + 1 + \frac{1}{K^2}\right) \quad (20)$$

is found asymptotically for $t \gg \tilde{\Delta}^{-1}$.

In the opposite limit of extremely short times, $t \ll \alpha/\tilde{v}$, i.e., for times shorter than the high energy time scale, however, we obtain an $x$ *independent* decay with an exponent

$$F_{RR}(x,t) \approx -\frac{2\gamma t}{\pi\alpha}, \qquad t \ll \alpha/\tilde{v}. \quad (21)$$

The richest behavior is found in the intermediate temporal region, $\alpha/\tilde{v} \ll t \ll 1/\tilde{\Delta}$, where the first term in Eq. (18) gives results identical to those in Eq. (20). The second term in Eq. (18) with $m_q^B(t)$ gives, however, a correction for $2\tilde{v}t \ll |x|$

$$\frac{G(x;t)}{G(x;0)} \approx e^{-\frac{\gamma t}{2K^2\tilde{\xi}}\left|\frac{x}{\tilde{\xi}}\right|}e^{-\frac{\gamma t\left(1+K^2\right)}{\pi\alpha K^2}} \times$$
$$\times \begin{cases} 1, & 2\tilde{v}t \gg |x|, \\ e^{-\frac{\gamma}{4\tilde{v}K}\left(K - 2\frac{\tilde{v}t|x|}{K\tilde{\xi}^2} - \frac{1}{K}\right)}, & 2\tilde{v}t \ll |x|, \end{cases} \quad (22)$$

exhibiting very mild light cone behavior at $2\tilde{v}t = x$, compared to the one found in closed quantum systems [33].

In the gapless case, $\Delta = 0$, Eq. (22) renders the results obtained for dissipative Luttinger liquids [27]. Notice, however, that the intricate spatial and temporal behavior of Eq. (22) is valid only for $\alpha/\tilde{v} \ll t \ll 1/\tilde{\Delta}$, and the late time, large distance behavior of the correlation function is dominated by the gap, giving rise to the unusual $\exp(-\text{const} \cdot |x|t)$ decay of $G_{RR}(x;t)$, one of the key findings of the present paper. This latter behavior starts to become dominant for $x \gtrsim \tilde{\xi}^2/\alpha$. This strong suppression of the correlation function at late times is in accord with what is expected from a very high temperature state, where the correlations are suppressed by the time and lengthscale associated to temperature[8].

We compare the analytical estimates above with the numerical evaluation of Eq. (18) in Fig. 1, where we show the effective decay rate, $\sim \frac{1}{t} \ln \left[ G_{RR}(x;t)/G_{RR}(x;t) \right]$ as a function of time. The spatial and temporal dependence of the single particle density matrix $G_{RR}(x;t)$ agrees well with the analytical results, confirming the $\exp(-\text{const} \cdot |x|t)$ asymptotics (horizontal dashed lines), while the early time decay rate remains largely insensitive

to the spatial separation. The temporal and spatial decay is partially confirmed by exact diagonalization simulation of the $XXZ$ Heisenberg model. Since exact diagonalization has high computational costs, only system sizes of up to 14 sites are accessible which is less conclusive for long distances.

## IV. CORRELATION BETWEEN LEFT- AND RIGHT MOVING PARTICLES

While in the Luttinger liquid phase, left- and right-movers are independent, they become correlated in the presence of a gap [48, 49], and the corresponding off diagonal density operator

$$G_{LR}(x;t) = \text{Tr}\left\{ \rho(t)\, \psi_L^+(x)\psi_R(0) \right\} \qquad (23)$$

becomes finite. The trace is evaluated by using the bosonic representation, Eq. (11), similarly to the previous section. The decay factor of the single particle density matrix is obtained as

$$
\begin{aligned}
F_{LR}(x;t) &\equiv \ln\left( \frac{G_{LR}(x;t)}{G_{LR}(x;0)} \right) = \qquad\qquad\qquad\qquad\qquad\qquad\qquad\qquad\qquad (24)\\
&= -\sum_{q>0} \frac{4\pi}{L|q|} \left( n_q^B(t)\frac{\omega(q) - g(q)\cos(qx)}{\tilde{\omega}(q)} + \text{Re}\left(m_q^B(t)\right)\frac{\omega(q)\cos(qx) - g(q)}{\tilde{\omega}(q)} - \text{Im}\left(m_q^B(t)\right)\sin(qx) \right),
\end{aligned}
$$

which, interestingly, depends on time again only through the functions $n_q^B(t)$ and $m_q^B(t)$. Details of the derivation can be found in the Appendix. The initial correlation function now reads

$$G_{LR}(x;0) = \frac{1}{2\pi\alpha} e^{-\sum_{q>0}\frac{2\pi}{L|q|}\left[\mathcal{K}(q) + \frac{g(q)}{\tilde{\omega}(q)}(1-\cos(qx))\right]} \quad (25)$$

and is evaluated as

$$G_{LR}(x;0) = \frac{1}{2\pi\alpha}\left(\frac{\alpha}{\tilde{\xi}}\right)^{\frac{1+K^2}{2K}}
\begin{cases}
\left|\frac{\tilde{\xi}}{x}\right|^{\frac{1-K^2}{2K}}, & \alpha \ll x \ll \tilde{\xi}, \\[2ex]
e^{-\frac{\pi}{4K}\left|\frac{x}{\tilde{\xi}}\right|}, & \alpha \ll \tilde{\xi} \ll x.
\end{cases}
\quad (26)$$

Note that if the gap is zero, then $\tilde{\xi}$ is infinitely large, and the correlation function vanishes. Without forward scattering, $K = 1$, $G_{LR}(x;0)$ stays constant within the coherence length, and vanishes exponentially outside it. The local density operator, $G_{LR}(0;t)$, characterizes the amplitude of the $2k_F$ oscillating charge density profile. At the start of the dissipation quench and in the $\alpha\tilde{\Delta} \ll \tilde{v}$ limit we have

$$G_{LR}(0;0) = \frac{1}{2\pi\alpha}\left(\frac{\alpha e^{\gamma_E}\tilde{\Delta}}{2\tilde{v}}\right)^K, \qquad (27)$$

with $\gamma_E = 0.5772\ldots$ denoting Euler's constant. In the absence of forward scattering interactions, $K = 1$, the charge density wave amplitude $\langle \psi_L^+(x)\psi_R(0)\rangle$ is directly proportional to the gap, $G_{LR}(0;0) \sim \Delta$, while this gets significantly renormalized due to interactions, $K \neq 1$, as $\Delta \to \tilde{\Delta}^K$. The exponent $K$ is inherited from the scaling dimension of operator responsible for $2k_F$ density fluctuations [8]. In particular, repulsive interactions with $K < 1$ favor ordering by enhancing the value of the gap and also the amplitude of charge oscillations, while attractive interactions with $K > 1$ reduce the gap, and suppress charge oscillations.

At long distances, the time-dependent part exhibits the same limiting behavior as the density matrix of the right movers,

$$F_{LR}(x,t) \approx
\begin{cases}
-\frac{\gamma t}{\pi\alpha}\left(\frac{\pi\alpha|x|}{2\tilde{\xi}^2 K^2} + 1 + \frac{1}{K^2}\right), & t \gg 1/\tilde{\Delta}, \\[2ex]
-2\frac{\gamma t}{\pi\alpha}, & t \ll \alpha/\tilde{v}.
\end{cases}
\quad (28)$$

For long times and large distances, the gap thus dominates and leads to a $G_{LR} \sim e^{-\text{const}\cdot|x|t}$ behavior, as also confirmed by the numerical evaluation of Eq. (24) (see Fig. 2).

At intermediate times, $\alpha/\tilde{v} \ll t \ll 1/\tilde{\Delta}$, the decay is

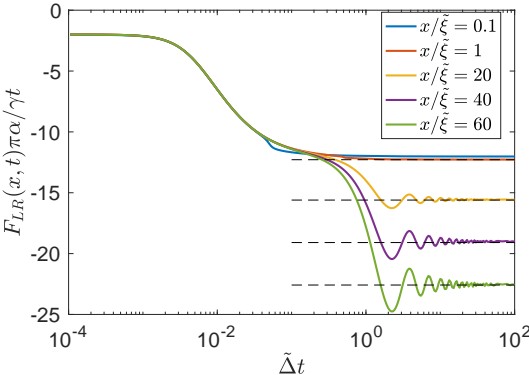

FIG. 2. Time-dependence of the decay rate of the anomalous $LR$ single particle density matrix, governing the amplitude of $2k_F$ charge density wave correlations. The curves are obtained by evaluating Eq. (24) numerically with $K = 0.3$ and $\alpha/\tilde{\xi} = 0.01$. Short and long time asymptotics are very similar to those of the $RR$ sector (see in Fig. 1). The crossover is described by Eq. (29).

slightly different than the one obtained in the $RR$ and $LL$ sectors. We obtain, in particular

$$\frac{G_{LR}(x;t)}{G_{LR}(x;0)} \approx e^{-\frac{\gamma t}{2K^2\tilde{\xi}}\left|\frac{x}{\tilde{\xi}}\right|} e^{-\frac{\gamma t\left(1+K^2\right)}{\pi\alpha K^2}} \times$$

$$\times \begin{cases} e^{\frac{\gamma}{2\tilde{v}}\left(\frac{\mathrm{sgn}(x)}{K}-1\right)}, & 2\tilde{v}t \gg |x|, \\ e^{\frac{\gamma}{2\tilde{v}}\left(\frac{1-K^2}{2K^2}+\frac{\tilde{v}tx}{K^2\tilde{\xi}^2}\right)}, & 2\tilde{v}t \ll |x|, \end{cases} \quad (29)$$

which again features rather mild light cone effects.

We can also follow the dissipation-induced temporal destruction of the $2k_F$ charge density wave. The amplitude $\langle\psi_L^+(0)\psi_R(0)\rangle$ decays exponentially in time as

$$\frac{G_{LR}(0;t)}{G_{LR}(0;0)} \approx \begin{cases} e^{-\frac{4\gamma t}{\pi\alpha}}, & t \ll \alpha/\tilde{v}, \\ e^{-\frac{2\gamma t}{\pi\alpha}-\frac{\gamma}{2\tilde{v}}}, & \alpha/\tilde{v} \ll t \ll 1/\tilde{\Delta}, \\ e^{-\frac{2\gamma t}{\pi\alpha}}, & 1/\tilde{\Delta} \ll t \end{cases} \quad (30)$$

with the decay rate reduced by a factor of 2 in course of the dissipative time evolution, as shown in Fig. 3.

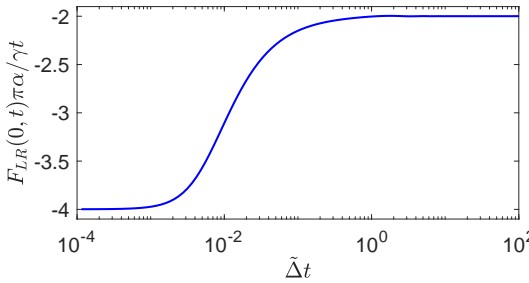

FIG. 3. Time-dependence of the decay rate of charge density wave order parameter for $\alpha\tilde{\Delta}/\tilde{v} = 0.01$, (see Eq. (30)). Apart from the renormalization of its equilibrium value $\tilde{\Delta} \to \tilde{\Delta}^K$, the time dependence is completely independent of the forward scattering interaction.

## V. ENTANGLEMENT ENTROPY GENERATION

The information loss in an open system is expressively demonstrated by the growth of the von Neumann entropy, defined as $S(t) = -\mathrm{Tr}\{\rho(t)\ln\rho(t)\}$. This latter can be regarded as the thermodynamical entropy, or as the measure of entanglement with the dissipative environment. Following the derivation of Ref. [27], the entropy is obtained as

$$S(t) = 2\sum_{q>0}\left[(N_q(t)+1)\ln(N_q(t)+1) - N_q(t)\ln N_q(t)\right] \quad (31)$$

where $N_q(t) = \sqrt{\left(n_q^B(t)+\frac{1}{2}\right)^2 - |m_q^B(t)|^2} - \frac{1}{2}$, with $n_q^B(t)$ and $m_q^B(t)$ the functions displayed in Eq. (9). For weak dissipation, i.e., when $\gamma \ll \tilde{v}K$, we get $N_q(t) \approx \gamma\tilde{\omega}(q)t/(\tilde{v}K\pi)$ and the entropy is obtained as

$$S(t) \approx \frac{L}{\pi\alpha} \begin{cases} -f\left(\alpha/\tilde{\xi}\right)\frac{\gamma t}{K\alpha\pi}\ln\frac{\gamma t}{K\alpha\pi}, & \gamma t \ll K\alpha\pi, \\ \ln\frac{\gamma t}{K\alpha\pi}, & \gamma t \gg K\alpha\pi \end{cases} \quad (32)$$

where $f(z) = \frac{\pi z}{2}\left(-Y_1(z)+\mathbf{H}_1(z)\right)$ with $Y_1$ the Bessel function of the second kind, and $\mathbf{H}_1$ is Struve function. In the scaling limit, we have $f(\alpha \ll \tilde{\xi}) = 1$, and the entropy exhibits the same time dependence as in the gapless case [27]. This behavior is explained by the fact that the entropy is mostly determined by high energy modes, and is not influenced by the presence or absence of the gap at low energies. We mention that for fermionic models with finite local Hilbert space, the entropy cannot be larger than $\sim L/\alpha$. Therefore, the second line in Eq. (32) could possibly show up in bosonic realization of the sine-Gordon models[7], where the local Hilbert space is much bigger.

Beside the von Neumann entropy, it is worth investigating the second Rényi entropy, which is more relevant experimentally [50–52]. The second Rényi entropy is defined as $S_2(t) = -\ln\mathrm{Tr}\{\rho(t)^2\}$, and is computed as

$$S_2(t) = 2\sum_{q>0}\ln\left(2N_q(t)+1\right). \quad (33)$$

Details of the derivation can be found in the Appendix. In the weak dissipation limit, the early and long time dependence is calculated as

$$S_2(t) = \frac{L}{\pi\alpha} \begin{cases} 2f\left(\frac{\alpha}{\xi}\right)\frac{\gamma t}{K\alpha\pi}, & \gamma t \ll K\alpha\pi, \\ \ln\frac{\gamma t}{K\alpha\pi}, & \gamma t \gg K\alpha\pi \end{cases} \quad (34)$$

which is again dominated by high energy modes and the gap does not affect the asymptotic temporal variations. Similar time dependencies are found in other systems as well[12].

We mention that the monotonic temporal increase of these entropies agrees with the physical picture that the system heats up to infinite temperatures upon coupling it

to a bath through a hermitian jump operator. However, the late time $\ln(t)$ entropy growth is connected to the infinite dimensional local Hilbert space of the $b_q$ bosons, and is absent in lattice models with a finite dimensional local Hilbert space (i.e. for fermions or hard core bosons). In this limit, the sine-Gordon description will eventually break down and the entropies are expected to saturate to their maximal values, determined by the size of the Hilbert space.

## VI. CONCLUSIONS

We investigated the dissipative dynamics of the sine-Gordon model in its 'semi-classical' limit, where forward scattering interactions are still incorporated, but solitons are neglected, and the low energy Hamiltonian is described in terms of massive bosons. We considered the case of a current operator induced dissipation within the Lindblad formalism, where the dissipative dynamics remains still exactly solvable. We have focused, in particular, on the time evolution of the density operator of the underlying fermionic theory.

The interplay of forward scattering interaction, gap, and dissipation produces interesting features in the single particle density matrix. The density matrix of the right movers features Luttinger liquid type power law correlations within the coherence length, associated with the gap, $|x| < \tilde{\xi}$, but decays exponentially in space and time as $\sim \exp(-\text{const} \cdot |x|t)$ in the long distance, late time limit.

More importantly, the anomalous density matrix between right *and* left moving fermions reveals a very similar spatio-temporal pattern. In the initial non-dissipative state, the bare gap $\Delta$ gets strongly renormalized by interactions, $\tilde{\Delta} \to \Delta^K$ with $K$ the Luttinger liquid parameter, and gets exponentially destroyed in time with a decay rate that gradually decreases by a factor of 2.

While correlations and the single particle density matrix are very sensitive to the presence of the gap $\Delta$, the entropy is not. The von-Neumann and second Rényi entropies are extensive and grow initially as $-t\ln(t)$ and $t$, respectively due to the presence of interaction with the dissipative environment. This originates from the large number of high energy modes, which become populated fast due to heating from dissipation.

## ACKNOWLEDGMENTS

This research has been supported by the National Research, Development and Innovation Office - NKFIH within the Quantum Technology National Excellence Program (Project No. 2017-1.2.1-NKP-2017-00001), research projects K119442 and K134437, and within the Quantum Information National Laboratory of Hungary, and by the Romanian National Authority for Scientific Research and Innovation, UEFISCDI, under project no. PN-III-P4-ID-PCE-2020-0277.

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

## Appendix A: Derivation of the equal-time single-particle density matrix

In this section, details of the derivation of the single-particle density matrix are provided in both the $RR$ and the $LR$ channel. In both cases, the time-dependence of the density matrix is necessary. The Lindbladian dynamics does not couple the different $q > 0$ modes and, hence,

the density matrix matrix can be written in the form of

$$\rho(t) = \prod_{q>0} \left( \frac{\nu_q(t)^2 - |c_q(t)|^2}{\nu_q(t) + 1} \times \right.$$

$$\left. \times e^{c_q(t)K_{q-}} e^{-2\ln(\nu_q(t)+1)K_{q0}} e^{c_q(t)^* K_{q+}} \right) \qquad (A1)$$

where $K_{q-} = b_q b_{-q} = K_{q+}^+$ and $K_{q0} = (b_q^+ b_q + b_{-q}^+ b_{-q})/2$ obey $su(1,1)$ algebra. By substituting (A1) into the Lindblad equation (8), first order, non-linear differential equations are derived for $\nu_q(t)$ and $c_q(t)$. As shown in Refs. [53], the differential equations are solved by

$$\nu_q(t) = \frac{n_q^b(t)}{n_q^b(t)^2 - |m_q^b(t)|^2} \qquad c_q(\tau) = \frac{m_q^b(t)}{n_q^b(t)^2 - |m_q^b(t)|^2} \qquad (A2)$$

where $n_q^b(t) = \mathrm{Tr}\left[\rho(t) b_q^+ b_q\right]$ and $m_q^b(t) = \mathrm{Tr}\left[\rho(t) b_q^+ b_{-q}^+\right]$. These expectation values are expressed in terms of $B$-bosons as given in Eq. (14) of the main text and

$$m_q^b(t) = \frac{\omega(q)}{\tilde{\omega}(q)} \mathrm{Re}\, m_q^B(t) + i\mathrm{Im}\, m_q^B(t) + \frac{g(q)}{\tilde{\omega}(q)} \left( \frac{1}{2} + n_q^B(t) \right) . \qquad (A3)$$

The equal-time single-particle density matrix of the $RR$ channel is defined as

$$G_{RR}(x;t) = \mathrm{Tr}\left\{ \rho(t)\, \psi_R^+(x)\psi_R(0) \right\} , \qquad (A4)$$

where $\psi_R(x)$ is the field operator of right moving fermions. By substituting its bosonized form and following standard steps [8, 54], we obtain

$$\ln(2\pi\alpha G_{RR}(x;t)) = \sum_{q>0} \frac{2\pi}{L|q|} \left( i\sin(qx) - \right.$$

$$\left. -(1 - \cos(qx)) \left( 1 + \frac{2\nu_q(t)}{\nu_q(t)^2 - |c_q(t)|^2} \right) \right) . \qquad (A5)$$

Using the relations of (A2) and recognizing the non-interacting single-particle density matrix in the time-independent terms, the expression is rewritten as

$$\ln G_{RR}(x;t) = \ln G_0(x) - \sum_{q>0} \frac{4\pi}{L|q|} n_q^b(t)(1 - \cos(qx)) \qquad (A6)$$

which is identical to Eq. (13) of the main text.

In the $LR$ sector, the equal-time single-particle density matrix is defined as

$$G_{LR}(x;t) = \mathrm{Tr}\left\{ \rho(t)\, \psi_L^+(x)\psi_R(0) \right\} \qquad (A7)$$

where $\psi_L(x)$ is the field operator of left-movers. By substituting the bosonized forms and following similar steps, the correlation function is calculated as

$$\ln(2\pi\alpha G_{LR}(x;t)) = -\sum_{q>0} \frac{2\pi}{L|q|} \left( 1 + 2n_q^b(t) + 2\mathrm{Re}\left( m_q^b(t)e^{iqx} \right) \right) . \qquad (A8)$$

Substituting (14) and (A3) leads to Eq. (24) in the main text.

### Appendix B: Derivation of the entropy

The second Rényi entropy is defined as $S_2(t) = -\ln\mathrm{Tr}\left\{\rho(t)^2\right\}$ where the density matrix is given in the form of (A1). For each wavenumber channel, the density operator can be expressed as a single exponential and the exponent may be diagonalized as $\rho_q(t) = (1 - e^{-\Omega_q(t)})^2 e^{-\Omega_q(t)(\beta_q^+(t)\beta_q(t) + \beta_{-q}^+(t)\beta_{-q}(t))}$ with some bosonic operator $\beta_q(t)$, see the Supplementary Material of Ref. [27], and

$$\Omega_q(t) = \left| \mathrm{acosh}\left( \frac{\nu_q(t)^2 - |c_q(t)|^2}{2(\nu_q(t)+1)} + 1 \right) \right| . \qquad (B1)$$

The function $\Omega_q(t)$ is also related to the instantaneous expectation value $N_q(t) = \langle \beta_q^+(t)\beta_q(t) \rangle$ by

$$N_q(t) = \frac{1}{e^{\Omega_q(t)} - 1} = \sqrt{\left( n_q^b(t) + \frac{1}{2} \right)^2 - |m_q^b(t)|^2} - \frac{1}{2} . \qquad (B2)$$

Interestingly, the latter formula does not change if we use $n^B$ and $m^B$ instead of $n^b$ and $m^b$. By substituting (14) and (A3), we obtain

$$N_q(t) = \sqrt{\left( n_q^B(t) + \frac{1}{2} \right)^2 - |m_q^B(t)|^2} - \frac{1}{2} . \qquad (B3)$$

The trace of the entropy is evaluated as

$$S_2(t) = 2\sum_{q>0} \ln\left( \frac{1 + e^{-\Omega_q(t)}}{1 - e^{-\Omega_q(t)}} \right) = 2\sum_{q>0} \ln\left( 1 + 2N_q(t) \right) \qquad (B4)$$

which is identical to Eq. (33) of the main text.

In the weak dissipation limit, i.e. when $\gamma \ll \pi\tilde{v}K$, the approximation of $N_q(t) \approx n_q^B(t)$ can be used.