# Peer review of "Dissipative dynamics in the free massive boson limit of the sine-Gordon model"

_SciPost Physics_

## Round 1 · Referee Report · Anonymous (Referee 1) · 2021-9-30

Report

The article studies the dissipative dynamics, implemented via a Lindblad-type of coupling to the current, of the one-dimensional Klein-Gordon model. The latter is frequently referred to as the massive boson limit of the sine-Gordon model; while this is technically correct is somewhat leaves the impression that typical features of the sine-Gordon theory are captured which, in my opinion, are not. In particular, the defining soliton excitations of the sine-Gordon model are absent in the considered approximation. It would be nice to comment on the expected effects of keeping the solitons in the dynamics.

Nevertheless, the manuscript addresses a relevant question, which, due to considering the massive boson limit, can be analysed using standard methods. In both the applied techniques and the considered setup the manuscript is a direct generalisation of Reference [25], which treated the same question in the massless limit of the Klein-Gordon model, namely the usual Luttinger theory. In fact, this reference provides the main result for comparison, which is performed in reasonable detail.

The manuscript is written in a clear way, containing sufficient details and references. Thus I conclude that it meets the general acceptance criteria of SciPost. Given that the manuscript is a direct generalisation of Reference [25], I am not sure whether it also satisfies the expectations like "above-the-norm degree of originality". I certainly do not believe that the work satisfies any of the acceptance criteria of SciPost Physics, like a "groundbreaking theoretical/experimental/computational discovery".

Thus while I cannot recommend publication in SciPost Physics, the manuscript may be suitable for a more specialised journal like SciPost Physics Core.

Requested changes

Please address the following points: 1. Is it possible to analyse the problem also using numerical approaches by, for example, considering the mentioned XXZ chain? 2. A discussion about the role of solitons should be added. Is there any expectation about their effect on the dynamics? What would be the technical problems when including solitons? 3. For the dynamics of closed sine-Gordon/Klein-Gordon systems the References [3,4] are given. Are these the only ones, or the most representative ones?

---

## Round 1 · Referee Report · Anonymous (Referee 2) · 2021-10-14

Strengths

1- Exact analytical solution of the problem and derivation of results 2- Detailed presentation of the behaviour of physical quantities

Weaknesses

1- Limited discussion of physical significance of the chosen protocol and interpretation of the results 2- Calculations not presented in full detail / replaced by references to earlier publications involving very similar calculations

Report

This work studies the dissipation dynamics of the sine-Gordon in a special limit, the free massive boson limit. Dissipation is accounted for in the form of the Lindblad equation where the jump operators are chosen as the currents of the noninteracting massless model. The initial state is the ground state of the closed system in the same special limit. The dynamical equations can in this case be solved exactly analytically using standard linear transformation techniques presented in earlier works studying technically similar problems ([25,41] by some of the same authors). The main results are expressions for the time evolution of densities, correlation functions and entropies, whose behaviour as functions of space and time is analysed in detail, extracting their scaling laws.

This work is expected to serve as a report of exact results on the increasingly studied subject of dissipation in a many-body model of significant interest. These should be useful for comparison in future numerical studies of the problem in the truly interacting regime of the model or in its many applications in condensed matter and cold-atom physics. The presentation of the paper is clear, especially the discussion of the scaling of physical quantities which is very detailed. Nevertheless in several places the justification of certain choices or steps of the calculation is not included in the text but delegated to the earlier works [25,41]. A weak point is also the absence of an in-depth discussion of the physical significance and interpretation of the results, even though I appreciate that this is partly due to that the Lindblad equation describes dissipation in somewhat abstract terms rather than referring to a concrete dissipation mechanism.

Based on the above and taking into account the acceptance criteria, I recommend publication in SciPost Physics Core instead of SciPost Physics. I would also ask the authors to take into account the following proposed changes.

Requested changes

1- The authors refer to the special limit of the sine-Gordon model they study as the 'massive boson' limit, but this is not a standard or unambiguous term: the sine-Gordon model is an interacting bosonic model and it's massive in a large part of its phase diagram, not only in this special limit. The term 'free' or 'quadratic massive boson' limit is more accurate.

2- As already mentioned, the main thing that is missing from this work is a discussion of the physical significance and interpretation of the results. Why is this choice of jump operators natural? How do the results match with expectations for a system that heats up? Even if some of this discussion is included in earlier works it is worth including and specialising it here. Also, given that the sine-Gordon model is only a low-energy approximation of the systems it is applied on and since the heating process results in higher and higher energy states, it is worth warning that in such systems the sine-Gordon description will eventually break down and the large time dissipative dynamics will be controlled by factors beyond that.

3- Including an appendix with a somewhat more detailed outline of the calculations and intermediate formulas would help to make the manuscript self-contained.

Minor points:

4- Motivation for the present work comes partly from Refs. [26, 27] which are about black hole evaporation / Hawking radiation. What is the connection to the problem of dissipation in the present settings?

5- Another motivation comes from the experimental observations of Ref. [28] and the theoretical study of Ref. [29]. It's worth however noting that some recent theoretical works (arXiv:2010.11214, arXiv:2012.05885) suggest a different explanation of that experiment (parabolic trap).

6- After Eq. (17): For short distances [...] the gap has no effect on the dynamics". There is no time involved in the expression this sentence refers to, it's a ground state equal-time correlator. So I think the authors meant to write something else not 'dynamics' here.

7- There are a few minor grammar errors or typos (e.g. missing articles, non-capital letters in titles in the biliography).

---

## Round 2 · Referee Report · Anonymous · 2021-11-24

Report

The authors have taken into account the remarks by the referees. Thus the manuscript can be published in its current form. However, given the acceptance criteria of SciPost Physics, my recommendation that the manuscript is suitable for a more specialised like SciPost Physics Core remains.

---

## Round 2 · Author Response

--- RESPONSE TO REFEREE REPORT 1 ---

We thank the Referee for the detailed report. Let us start by noting that to the best of our knowledge, it is not clear whether the dissipative sine-Gordon model is exactly solvable or not, unlike its non-dissipative counterpart, which is Bethe Ansatz solvable. Therefore, in this regard, any solution, which is reliable for a certain range of parameters, should be of high value. This is especially true when correlation functions can also be calculated. In this regard, our work represents the first step in this direction to understand the dynamics of the dissipative sine-Gordon model.

As to the comments of the Referee:

  1. We agree with the Referee that in principle it would be possible to study the dynamics, we predict, also numerically on the XXZ chain. However, given the nature of the various crossovers and associated lengthscales in the correlation functions, it would be very demanding to catch and identify all these features. We have performed exact diagonalization studies on the XXZ chain up to N=14 sites and found partial agreement with our analytical results. However, due to the small system sizes, these data were inconclusive, therefore we refrain from displaying it in the paper.

  2. As to the effect of the solitons, as long as the Luttinger liquid parameter is small, i.e. K<<1, expanding the cosine potential around one of its minima is a rather good approximation and one does not need to worry about tunneling to adjacent minima, as also mentioned in the newly cited papers by Foini et al. and Ruggiero et al. In this limit, the soliton mass is pushed up to very high energies with 1/\sqrt(K) and the solitons practically decouple from the dynamics. However, their effect can most probably be taken into account using a form factor expansion or by using a perturbative expansion in $K$, which is only expected to affect the short time transient dynamics.

  3. The references we cite in connection to the sine-Gordon model are certainly not the most ideal papers, so we added some additional references.

--- RESPONSE TO REFEREE REPORT 2 ---

We thank the Referee for his/her detailed report. Before we respond to each requested change, let us emphasize that to the best of our knowledge, it is not clear whether the dissipative sine-Gordon model is exactly solvable or not, unlike its non-dissipative counterpart, which is Bethe Ansatz solvable. Therefore, in this regard, any solution, which is reliable for a certain range of parameters, should be of high value. This is especially true when correlation functions can also be calculated. In this regard, our work represents the first step in this direction to understand the dynamics of the dissipative sine-Gordon model.

  1. We thank for raising our attention to this unambiguousness. We changed the terminology accordingly.

  2. Using the local current as jump operator is rather natural since it can easily arise from fluctuating gauge fields or vector potentials, which couple naturally to the current. Another natural choice for the jump operator would be the local density, and our calculations can be applied for that case as well with minor modifications, e.g. by replacing K with 1/K. We also explained that strong spatio-temporal suppression of correlation function is in accord with what is expected from a very high temperature thermal state where correlations are suppressed beyond the time and length scale associated to temperature. In connection to the entropies, we mention that their monotonic temporal increase agrees with the physical picture that the system heats up to infinite temperatures upon coupling it to a bath through a hermitian jump operator. However, the late time $\ln(t)$ entropy growth is connected to the infinite dimensional local Hilbert space of the $b_q$ bosons, and is absent in lattice models with a finite dimensional local Hilbert space (i.e. for fermions or hard core bosons). In this limit, the sine-Gordon description will eventually break down and the entropies are expected to saturate to their maximal values, determined by the finite size of the Hilbert space.

  3. A new appendix is added, where the main steps for calculating the correlations functions and the entropies are highlighted.

  4. The Hawking radiation is presumably not the most ideal motivation for this study, we replaced it by "targeted cooling into topological states from arbitrary initial states".

  5. We added these two citations.

  6. We thank for the remark and corrected the expression accordingly.

  7. We tried to correct the reference list. Some issues must be related to some bibtex setting, everything looks perfect in our .bib file. This will certainly be fixed in the published version.

---

## Round 2 · List of Changes

- The terminology of the limit is clarified as "free massive boson limit of the sine-Gordon model" throughout the whole manuscript including the title.
- In the second paragraph of the introduction, targeted cooling is added as related phenomenon, new references are also cited.
- Between Eqs. (5) and (6), we added a new paragraph about expected effect of solitons.
- Before Eq. (7), explanation is added about the choice of jump operators.
- Eq. (9a) is reformulated.
- New appendices are added to explain the derivation of Eqs. (13), (24) and (33).
- The sentence after Eq. (17) is corrected.
- Additional interpretation of the results is added to the end of the second last paragraph of Sec. III.
- Remark about numerical simulation is added to the last paragraph of Sec. III.
- Additional paragraph to the end of Sec. IV.

You are currently on this page

Resubmission 2108.05865v2 on 24 November 2021

---

## Editorial Decision

editor-in-charge_assigned